# Endogenous Retroviruses as Modulators of Innate Immunity

**DOI:** 10.3390/pathogens12020162

**Published:** 2023-01-19

**Authors:** Eric Russ, Sergey Iordanskiy

**Affiliations:** 1Department of Pharmacology & Molecular Therapeutics, Uniformed Services University of the Health Sciences, Bethesda, MD 20814, USA; 2The Henry M. Jackson Foundation for the Advancement of Military Medicine, Bethesda, MD 20817, USA; 3Graduate Program of Cellular and Molecular Biology, Uniformed Services University of the Health Sciences, Bethesda, MD 20814, USA; 4Armed Forces Radiobiology Research Institute, Uniformed Services University of the Health Sciences, Bethesda, MD 20814, USA

**Keywords:** human endogenous retroviruses (HERVs), innate immunity, provirus, interferon, pattern-recognition receptors (PRR), RIG-I-like receptors (RLR), cytokines

## Abstract

Endogenous retroviruses (ERVs), or LTR retrotransposons, are a class of transposable elements that are highly represented in mammalian genomes. Human ERVs (HERVs) make up roughly 8.3% of the genome and over the course of evolution, HERV elements underwent positive selection and accrued mutations that rendered them non-infectious; thereby, the genome could co-opt them into constructive roles with important biological functions. In the past two decades, with the help of advances in sequencing technology, ERVs are increasingly considered to be important components of the innate immune response. While typically silenced, expression of HERVs can be induced in response to traumatic, toxic, or infection-related stress, leading to a buildup of viral transcripts and under certain circumstances, proteins, including functionally active reverse transcriptase and viral envelopes. The biological activity of HERVs in the context of the innate immune response can be based on the functional effect of four major viral components: (1) HERV LTRs, (2) HERV-derived RNAs, (3) HERV-derived RNA:DNA duplexes and cDNA, and (4) HERV-derived proteins and ribonucleoprotein complexes. In this review, we will discuss the implications of HERVs in all four contexts in relation to innate immunity and their association with various pathological disease states.

## 1. Introduction

Transposable elements (TEs) comprise nearly half of the human genome (~45%) and are segments of DNA that move within the genome in either a conserved (the number of copies stays the same) or replicative (the number of copies increases by at least one) manner [1]. Based on the mechanism of movement, otherwise known as transposition, TEs are divided into two broad classes: (1) DNA transposons and (2) retrotransposons or retroelements. DNA transposons comprise 2.8% of the human genome and move via a “cut-and-paste” (conservative transposition) mechanism, where the TE undergoes excision from its original position in the genome and integration into a different location [1]. Conversely, retroelements comprise 42.2% of the human genome and move via a “copy-and-paste” (replicative transposition) mechanism that involves transcription of the TE to produce a copy in the form of an RNA intermediate, followed by reverse transcription and integration into the genome [1].

Retroelements are further divided into two groups based on the presence of long-terminal repeats (LTRs) flanking the coding region of the retroelement: the non-LTR retroelements (33.9% of the human genome) and the LTR retroelements (8.3%) [1]. The non-LTR retroelements consist of short interspersed nuclear elements (SINEs), long interspersed nuclear elements (LINEs), and processed pseudogenes. Although their mechanism of transposition is reminiscent of retroviruses, they lack an LTR and take the appearance of integrated mRNA. On the other hand, the LTR retroelements have a similar DNA sequence topology as exogenous retroviruses and consist of endogenous retroviruses (ERVs) and elements of ERV origin, such as SINE-R. While DNA transposons and non-LTR retroelements are important components of the human genome, this review will primarily focus on ERVs as a “viral” component of the human genome, which, as we will discuss, can induce an immune response identical to the response to products of viral replication.

## 2. Origins of Endogenous Retroviruses

Retroviruses (family *Retroviridae*) are a diverse family of enveloped viruses that carry their genetic information in the form of positive single-stranded RNA [2]. Upon infection, the viral RNA genome is reverse transcribed into cDNA and permanently integrated into the host’s genome as a provirus. The provirus contains all of the necessary genetic components for viral replication, including an internal region encoding the essential viral genes *gag*, *pro*, *pol*, and *env*, followed by two flanking external regions that contain identical LTRs at the time of integration [2].

The *gag* gene encodes the major structural polyprotein, Gag, which forms the core of the virion [2]. The core is critical because it consists of the necessary viral proteins, incorporates two copies of the viral genomic RNA, and contains a tRNA that corresponds to the primer-binding site (PBS), a stretch of 18 nucleotides that prime reverse transcription of the viral RNA following entry of the target cell. The *pro* gene encodes for a viral protease that is essential for proteolytic processing of the viral polyproteins and may be contained within the core for post-entry steps of infection. The core also contains the products of the *pol* gene: reverse transcriptase to convert the RNA viral genome into cDNA, RNAse H to destroy the template RNA and assist in reverse transcription, and integrase to integrate the viral DNA genome into the target cell’s genome. Once assembled with its cargo, the core associates with the envelope protein(s), encoded by the *env* gene, at the cell surface to undergo viral budding and to facilitate the infection of new target cells. Finally, the flanking 5′ and 3′ LTRs contain the necessary regulatory elements, such as enhancers and promoters to recruit the required host cellular enzymes for transcription, the transcription state site (TSS) necessary to produce copies of the viral genome and to produce transcripts to be translated into the aforementioned viral proteins, the transcription termination signal to terminate transcription at the appropriate site, and the polyadenylation signal to enhance transcript stability and maturation [2].

Typically, spread of the retroviral genome from an original host into a new host occurs via horizontal transmission. This involves transcription of the provirus to produce an RNA genome, packaging of the genomic RNA molecules into a viral particle, release of the immature virion from the original host, maturation that involves proteolytic processing of viral protein-precursors by the viral protease, and uptake of the viral particle by a new host. However, in the event of a germline infection, the viral genome can transmit without viral particle production in a Mendelian fashion from parent to child, termed vertical transmission [3,4,5]. Retroviruses that transmit via vertical transmission as part of the host genome are termed ERVs and do not have the same dependency on the viral life cycle as exogenous retroviruses.

In the case of ERVs, since the DNA version of the viral genome is present in the original cell of the offspring, and by extension every nucleated diploid cell, the need for effective viral particle formation, secretion, and infection for viral propagation is removed. As a result, there is a lack of selective pressure for maintenance of the proviral genome and inactivating mutations accrue over time. Similar to the positive selection that host genes experienced throughout evolution, beneficial mutations of the ERVs are favored and their frequency within the population may grow to the point where it is fixed in the species. The most commonly known examples of this are syncytin 1 and 2, genes derived from the *env* genes of ERVW-1 and ERVFRD-1, respectively [6,7]. These ERV-derived genes produce proteins which perform functions similar to that of retroviral envelope proteins, such as cell fusion and immunomodulation, and it is suspected that higher mammals evolved to be dependent on these proteins for placentation [6,7]. However, in approximately 90% of the time, positive selection of an ERV provirus results in homologous recombination of the flanking LTRs, leaving behind only a solitary (“solo”) LTR [8,9] (Figure 1). For the proviruses that retained their internal region, the mutations accrued over time have typically rendered them non-protein coding and non-infectious, at least in humans [10]. Some proviruses may retain partial protein-coding capacity, particularly in the *env* region; however, no examples of replication-competent ERV proviruses have been identified in humans, but there are a few examples of viral-like particles that may form in abnormal conditions such as cancer [10,11,12,13].

Although most HERVs are suspected to be 20 to 40 million years old, ERV integration into the human lineage has occurred for at least 100 million years [10,14,15]. The youngest HERVs are those belonging to the HERV-K (HML-2) subfamily, with integration events ranging between 200,000 and 35 million years ago [16,17,18]. Since the HML-2 subfamily is relatively young compared to other HERV clades, they spent the least amount of time accruing mutations. This has allowed the HML-2 subfamily to have the highest degree of coding competence, including multiple intact open reading frames for each viral gene (*gag*, *pro*, *pol*, and *env*) and over 60 proviral loci with full-length or near full-length sequences [8]. From an evolutionary standpoint, specific HML-2 loci can be used to track human divergence and speciation, with approximately 120 sites (including both solo LTRs and proviral loci) present in modern humans and absent from chimpanzees, bonobos, and gorillas [19]. Interestingly, there are even differences in the HML-2 loci present in Neanderthals, Denisovans, and modern humans [20]. However, it is possible that the archaic hominid-specific loci exist in modern humans at a very low frequency and therefore have escaped detection, as approximately 10% of all HML-2 loci are polymorphic and therefore not fixed [21]. Nonetheless, the HML-2 subfamily clearly offers insight into human-specific integrations that may have shaped our evolution and divergence from our recent ancestors.

The existing nomenclature for HERVs is not entirely consistent, even following multiple attempts at creating a standard naming convention [22,23]. In general, HERVs are divided into three classes based on their similarity to exogenous retroviruses. Class I are *gammaretrovirus*- and *epsilonretrovirus*-like, class II are *betaretrovirus*-like, and class III are *spumaretrovirus*-like. Within these classes, there can be various families or subfamilies. Most families are named after the tRNA that corresponds to the PBS. For instance, the HERV-K family name implies that proviruses of this family use a lysine tRNA to prime reverse transcription. However, this naming convention is not always accurate, as some members of the HERV-K (HML-5) subfamily are suggested to use a methionine tRNA to prime reverse transcription, not lysine as the family name would imply [24]. For individual HERV proviral loci, the most common annotation is simply the HERV family they belong to, followed by a number or the cytogenetic location that the provirus maps to (“ERVK-5” and “HERV-K 3q12.3” both refer to a specific provirus belonging to the HERV-K family). In some cases, HERVs were named after the nearest gene (HERV-ADP) or after an amino acid motif (HERV-FRD). In total, there are 504 groups and over 700,000 HERVs have been identified [25]. For a comprehensive and up-to-date annotation of retroelements, Repbase provides a well-curated and regularly updated database for eukaryote repetitive sequences [26].

## 3. Epigenetic Regulation of HERV Activity

Before discussing the role of HERV elements as activators of the innate immune response, it is important to briefly review the mechanisms involved with silencing HERV activity under basal conditions. To prevent the unregulated activation of HERV LTRs, various epigenetic measures are implemented that largely revolve around altering the chromatin structure into a tightly packed state (also known as heterochromatin) through either DNA methylation or histone modifications (methylation and deacetylation) (Figure 2). However, since these modifications are not permanent, the epigenetic landscape of HERVs is not entirely conserved or consistent. Differences exist in the pattern of HERV expression from tissue-to-tissue and can vary depending on the immunological context, suggesting that the evolutionary framework was designed to regulate, not eliminate, HERV activity [27,28,29,30].

### 3.1. DNA Methylation

DNA modification by CpG methylation is carried out by DNA methyltransferases (DNMTs) that transfer a methyl group from S-adenyl methionine (SAM) to the fifth carbon of a cytosine residue (typically, one that precedes guanine) to form 5-methylcytosine (5mC) [31]. This is thought to result in transcriptional repression through two main mechanisms. One mechanism is through the direct interference of a transcription factor being able to bind to a transcription factor binding site (TFBS) due to presence of a nearby CpG methylation [32,33]. A second mechanism is through the recruitment of proteins with methyl-binding domains (MBDs) that induce histone deacetylation and subsequently, chromatin condensation, as discussed later [34].

Multiple studies have demonstrated a dependency on HERV methylation for their transcriptional repression [35,36,37]. Szpakowski et al. performed a genome-wide microarray analysis examining the presence of DNA methylation in HERV elements and its alteration in head and neck (HNC) cancer tissues versus adjacent non-tumor samples [35]. The HNC samples exhibited significantly lower levels of CpG methylation in various HERV elements and this loss had a significant correlation with HERV upregulation compared to adjacent non-tumor samples [35]. Interestingly, HERV methylation correlates with the age and length of the provirus, with younger and more intact HERV proviruses experiencing significantly higher levels of methylation and loss thereof following tumorigenesis. This suggests that DNA methylation-mediated HERV silencing may be more specific to young and intact proviruses, possibly due to the more viral-like nature of these proviruses compared to heavily mutated/inactive elements that might not require epigenetic silencing.

To examine the direct effect of CpG methylation on HERV activity, a DNMT inhibitor (DNMTi) can be used. Chiappinelli et al. demonstrated that DNMTi treatment with 5-azacytidine (Aza) or 5-aza-2′-deoxycytidine (Dac) induces significant HERV upregulation in multiple cell types, resulting in dsRNA accumulation and dsRNA sensing pathway activation [36]. Likewise, Roulois et al. had similar results with 5-AZA-CdR treatment in colorectal cancer-initiating cells (CICs), leading to the upregulation of at least 10 ERV subgroups and an enrichment in their dsRNA to ssRNA ratio [37]. Overall, the reduced level of DNA methylation and thereby elevated HERV transcription resulted in the activation of the RIG-I/MDA5/MAVS pathway (this pathway is discussed in more detail in Section 5 and the upregulation of interferon stimulated genes (ISGs).

### 3.2. Histone Modifications

The primary types of histone modifications that mediate transcriptional regulation are methylation and acetylation [38,39]. Histone methylation can occur on all three basic amino acid residues (lysine, arginine, and histidine) through the transfer of a methyl group from SAM to the ε-amino group of the targeted basic residue, with different possible amounts of methylation (mono-, di-, and tri-methylation) depending on the amino acid. Unlike CpG methylation, the impact of histone methylation on transcriptional activity is context dependent. Mechanistically, histone methylation either results in an electrostatic increase or decrease in the attraction between the histone and the surrounding DNA, thereby either inhibiting or enhancing the ability of transcription factors and RNA polymerase to access the DNA, respectively [40]. Two common examples of this are the H3K4me3 mark, which generally indicates open chromatin and active gene expression (or genes that are primed for expression), and the H3K27me3 mark, which is associated with compact chromatin and gene repression [41,42,43,44].

Several research groups have independently demonstrated a critical role for histone methylation in the silencing of ERVs during early embryo and germline development [45,46,47,48]. The major target of histone methylation in these cases was H3K9, which can be mediated by the methyltransferase G9A, among others. This led Liu et al. to explore the ability of a G9A inhibitor (G9Ai), UNC0638, to induce HERV expression in various human ovarian cancer cell lines [49]. They found that G9Ai treatment induced a distinct set of HERVs compared to DNMTi treatment. While DNMTi treatment activated relatively young HERVs that coincided with a high CpG density (in agreement with Chiappinelli et al. [35]), G9Ai treatment activated “middle-age” HERVs that had a low CpG density. These findings indicate that different epigenetic mechanisms work in tandem in order to target and silence HERV elements with differening sequence motifs and characteristics (such as CpG density). Following this, it was unsurprising that a combination DNMT inhibition with histone methyltransferase inhibition leads to a synergistic effect and increased tumor cell death, likely due to the higher abundance of HERV transcripts and consequently, increased viral mimicry and innate immune activation (discussed in Section 5) [49].

Another histone methyltransferase, SETDB1, can associate with KAP1 (also known as TRIM28) and sequence-specific zinc finger proteins (ZFPs) to mediate H3K9 trimethylation. While earlier reports suggested that ERV-associated histone methylation is not required for ERV silencing beyond early development, depletion of SETDB1 or TRIM28 in cancer cell lines contradicted these claims and indicated a positive role for histone methylation in the repression of HERVs outside of early development [50,51,52,53]. In support of this, Ceuller et al. identified SETDB1 as a critical regulator of leukemic cell survival out of a CRISPR-Cas9 screen of approximately 350 genes in THP-1 cells (a monocytic cell line), primarily due its role in repressing HERV-mediated activation of RIG-I/MDA5/MAVS signaling [50]. Specifically, RNA-sequencing identified that SETDB1 knockout resulted in the upregulation of numerous HERV families, including HERVL18, HERVP71A, HERV3-I, and more. HERV upregulation occurred bidirectionally and coincided with the induction of an anti-viral response, type I interferon signaling, and ISG expression, which could be abrogated through knockout of either the ssRNA receptor, RIG-I, or the dsRNA receptor, MDA5, suggesting that SETDB1 is required to prevent aberrant activation of the RIG-I/MDA5/MAVS signaling pathway. However, the effects of other retroelements, such as Alu and Line elements, were not ruled out and may activate this pathway in addition to HERVs.

Histone acetylation is regulated by two major forces: histone acetyltransferases (HATs) and histone deacetylases (HDACs). HATs transfer an acetyl group from acetyl CoA to the ε-amino group of lysine, thereby neutralizing the positive charge of the lysine residue and weakening the histone–DNA interaction. This results in an increase in accessibility for genes/elements contained within that region. HDACs do the opposite by removing the acetyl group from the lysine residue and thereby increase the histone–DNA interaction. Several studies have demonstrated that HDACi treatment induces HERV expression [54,55]. Curty et al. analyzed primary human CD4+ T-cells and their response to HDACi treatment with SAHA using RNA-sequencing. HDACi treatment over a 24 h period significantly modulated 722 HERVs, with the top 5 most upregulated HERV loci all containing open reading frames that could potentially allow for protein coding. Those HERV loci belonged to the families ERV3, HERV-W, MER4, and ERVL [54]. A similar study by White et al. demonstrated that the HDACi, vorinostat, modulated over 2000 individual HERV elements [55].

A general overview of the epigenetic regulation that HERVs experience is depicted in Figure 2. It is important to note, while an open chromatin state is critical for accessibility of the HERV to transcription factors and other transcriptional machinery, it is not the only factor involved in mediating HERV expression. If the required transcription factor (s) are not readily available in the nucleus or if the promoter region does not contain any functional TFBSs, then expression will remain silenced or limited [56,57,58].

## 4. HERV Elements as Genomic Enhancers of Innate Immunity

The co-option of HERVs during evolution has been proposed as a likely and beneficial route of facilitating the progression of the genomic regulatory network and for the improvement of pathway-specific gene expression [59,60]. This idea is based on several lines of reasoning and evidence. First, the ability of HERVs to move and amplify themselves within the genome combined with the regulatory elements contained within HERV LTRs allows for rapid changes in the genomic landscape that significantly outpaces the rate of random mutations alone [17]. Second, the LTR region can be adapted to benefit the host by regulating host gene expression in at least five different ways: (1) serve as an enhancer or repressor site, (2) serve as the promoter site, (3) provide the polyadenylation signal to terminate read-through transcripts, (4) provide a splice site, and (5) provide a source of RNA interference [25,61,62,63]. Third, selective dispersion of HERVs from the same family enables the recruitment of numerous genes into a specific regulatory network by providing an identical cis-acting regulatory element(s). Lastly, genomic studies have revealed the presence of over 790,000 HERV-derived TFBSs and over 150,000 HERV-derived DNase I hypersensitivity sites (DHS), indicating that HERV elements may be involved in the regulation of thousands of genes [64,65,66].

To broadly assess the capability of HERV-derived regulatory elements to bind specific transcription factors (TFs), Ito et al. performed a large-scale ChIP-seq analysis of 97 TFs and identified that HERV LTRs act as TFBSs for a wide array of TFs, including pluripotent TFs (SOX2, POU5F1, etc.), embryonic endoderm and mesendoderm TFs (GATA4/6, SOX17, etc.), hematopoietic TFs (SPI1, GATA1/2, etc.), and more [66]. An interesting finding was that LTRs with TF-bound regulatory elements were commonly located within close proximity to genes involved in various immune responses, especially those related to interferon signaling [66].

A separate study conducted by Chuong et al. more deeply explored the relationship between HERV elements and interferon signaling [65]. They initially probed for specific HERV elements that act as binding sites for STAT1 and IRF1, the major interferon γ-induced transcription factors, through ChIP-sequencing in two cell lines (K562 and HeLa), and primary CD14+ macrophages following interferon γ treatment [65]. Similar to Ito et al. [66], they found that HERVs bound by STAT1 and/or IRF1 were strongly enriched near interferon-stimulated genes (ISGs) and genes with an annotated immune function. To determine if there is a causal effect of HERV elements on nearby innate immunity genes, the authors sought to identify a specific sequence that was responsible for STAT1/IRF1 binding in a major HERV family that was enriched in their analysis. A highly represented HERV family bound to STAT1/IRF1 was MER41, consisting of six subfamilies (MER41A-G) derived from an endogenized gammaretrovirus that invaded human ancestors 45 to 60 million years ago. Sequencing analysis identified a STAT1 binding motif in the ancestral MER41 genome that is present in the consensus MER41B sequence but not in the consensus MER41A sequence due to a 43 bp deletion. Unsurprisingly, although their sequence homology is nearly identical aside from the 43 bp deletion, MER41B elements but not MER41A elements were enriched in the STAT1 bound sequence analysis. This suggested that the identified nucleotide sequence within MER41B represented a STAT1 binding site. Knockout of this motif using the CRISPR-Cas9 technique was performed with two guide RNAs that tightly surrounded the STAT1 binding motif of MER41.AIM2, a MER41B element located 220 bp upstream of the absent in melanoma 2 (AIM2) gene [65]. Importantly, AIM2 is a sensor of cytosolic dsDNA that forms a caspase-1 activating inflammasome to enhance innate immunity [67]. Since MER41.AIM2 was identified as the only STAT1 binding site within 50 kb of the AIM2 gene, it was predictable that AIM2′s upregulation following interferon γ treatment or infection by vaccinia virus was abolished in MER41.AIM2-KO cells. Phenotypically, caspase-1 levels were significantly reduced in the infected MER41.AIM2 knockout cells compared to their wild-type cell counterparts. Several other IFN-inducible genes demonstrated a similar, albeit not as strong, reliance on upstream MER41 elements for their induction following IFN signaling, including the ISGs APOL1, IFI6, and SECTM1 [65]. The reliance on MER41.AIM2 for AIM2′s upregulation is depicted in Figure 3, which can be used as a reference for other HERV elements that act as critical TFBSs for nearby innate immune-related genes [65,68,69].

Although the described knockout experiments are just a few examples of genes related to the innate immunity being dependent on HERV regulatory elements, this line of evidence in conjunction with the large-scale ChIP-seq analyses conducted by Chuong et al. and Ito et al. reinforce the notion that HERVs act as enhancers of genes involved in our innate immune response [65,66].

## 5. HERVs as Activators of the Innate Immune Response

Innate immunity is one of the most ancient lines of defense against infection and is distinguished from adaptive immunity through its ability to recognize a conserve set of molecular signatures to activate an immune response. The innate immunity sensors are termed pattern recognition receptors (PRRs) and recognize the presence of pathogen-associated molecular patterns (PAMPs) to signal infection and damage-associated molecular patterns (DAMPs) to signal cellular stress or damage. Most PRRs in vertebrate biology can be groups into five major classes: (I) toll-like receptors (TLRs), (II) retinoic acid-inducible (RIG)-I-like receptors (RLRs), (III) C-type lectin receptors (CLRs), (IV) nucleotide oligomerization domain (NOD)-like receptors (NLRs), and V) absent in melanoma-2 (AIM2)-like receptors (ALRs) [70]. An important PRR which does not fall into these five major groups is cyclic GMP–AMP synthase (cGAS), due to its relatively unique mechanism of inducing downstream signaling (as discussed later) [71,72]. While not all of these receptor classes have demonstrated the ability to recognize HERV-derived products, the viral-like nature of particular HERV-derived DNA, RNA, and proteins allows them to be recognized by and stimulate various PRRs to induce an innate immune response, similar to that of exogenous viruses. As a result, the effects of aberrant HERV upregulation are said to mimic the anti-viral state that is induced following viral infection, leading to coinage of the term “*viral mimicry*” when discussing the implications of HERV re-activation on the innate immune response [37,73,74]. Specific PRRs involved in the activation of innate immunity following HERV upregulation and their signaling pathways are depicted in Figure 4 and summarized in Table 1.

### 5.1. Toll-Like Receptors

TLRs are the first class of PRRs to be identified and consist of TLRs 1-10 in humans, although other TLRs have been identified and characterized in other species [70,82]. TLRs are type I transmembrane receptors that localize to the cell surface (TLRs 1, 2, 4, 5, 6, and 10) or to endosomal compartments (3, 7, 8, and 9), with the precise localization depending on the cell type and receptor specificity [83,84,85]. All TLRs have a similar structural arrangement that includes an N-terminal domain (NTD) responsible for PAMP/DAMP recognition, a transmembrane domain responsible for embedding the TLR within the cell surface or endosomal membrane, and a C-terminal domain (CTD) within the cytoplasm responsible for signal transduction through a toll-IL-1 receptor (TIR) homologous domain that recruits signal transduction adapter proteins. The NTD and CTD can also be called the ectodomain and endodomain, respectively. Upon ligand recognition, TLRs undergo dimerization and signal transduction to induce the activation of two distinct pathways: the MyD88-dependent pathway and the TRIF-dependent pathway. The first pathway is induced by all TLRs except TLR3 and activates NF-κB to induce NF-κB-associated pro-inflammatory gene and cytokine expression. The TRIF-dependent pathway activation is unique to TLRs 3 and 4 and activates IRFs 3 and 7 to induce type I interferons (IFN-I) and hence ISG expression.

While previous research has primarily focused on the role of TLRs in recognizing exogenous viruses, it is possible for HERVs to produce the general products for every virus-associated TLR (TLRs 3, 7, 8, and 9). To start off, TLR3 detects the presence of dsRNA and is expressed by cells of the central nervous system and the immune system [86]. Due to the flanking LTRs present in HERV proviruses, HERVs can be expressed bidirectionally and produce dsRNA complexes [36,87]. This provided a foundation for the belief that HERV dsRNA can agonize TLR3, which was eventually confirmed by Chiappinelli et al. [36]. They found that bidirectional ERV expression following DNMTi treatment activated TLR3 in HT-29 cells (human colorectal adenocarcinoma cell line) and subsequently promoted IRF7 phosphorylation and translocation into the nucleus to induce a robust type IFN-I response, including ISG expression. Additionally, the ability of human TLR3 to bind HERV RNA was also demonstrated by our group in THP-1 cells following exposure to gamma radiation, similarly leading to the induction of type I IFNs [75]. However, other pathways besides TLR3 may have been involved, such as the RIGI-MDA5-MAVS pathway, as discussed later in this section.

Intriguingly, although TLR3 is known for detecting dsRNA, Wang et al. showed that syncytin-1 protein, not dsRNA, can interact directly with TLR3 to induce downstream signaling and IRF3 phosphorylation, leading to IL6 expression [76]. In turn, TLR3-induced IL6 was shown to trigger the upregulation of C-reactive protein (CRP), a potent pro-inflammatory cytokine commonly found at higher levels in patients with a neurological disorder [88,89]. While it is difficult to prove, these results may explain the correlation between elevated syncytin-1 and CRP expression in patients with schizophrenia and other neurodegenerative diseases [76,90,91].

TLR4 is expressed in myeloid cells and is most well known for its ability to detect lipopolysaccharide (LPS), a component of nearly all gram-negative bacteria [92]. However, TLR4 is also known to recognize viral products, including HERV-derived proteins [93,94,95]. Initially identified in 1997 as a retrovirus detected in multiple sclerosis (MS) patients, multiple sclerosis-associated retroviral element (MSRV) was eventually acknowledged as an endogenous retrovirus belonging to the HERV-W family and its expression was shown to correlate with the inflammatory cytokines IL-6 and IL12p40 in MS patients [96,97]. It was not until 2006 that a mechanistic explanation for this correlation became available, with Rolland et al. demonstrating that the surface subunit (SU) of the Env protein derived from MSRV interacts with TLR4 and CD14 (a TLR4 co-receptor) to induce TLR4 signal transduction [77]. Ultimately, this Env–TLR4–CD14 interaction stimulates the production of multiple pro-inflammatory cytokines, such as TNF-a, IL1β, IL-12p40, IL-12p70, and IL6 and pro-inflammatory cell surface proteins (CD80, CD86, CD40, and HLA-DR) in human myeloid cell lineages and promotes the development of a Th1 type immune response [77]. Following this discovery, MSRV-derived Env and other HERV-W Env proteins have been shown to interact with TLR4 to induce a pro-inflammatory response in a variety of cell-types and situations, including a potential role in the development of type I diabetes (T1D) [98,99,100,101,102].

TLRs 7 and 8 sense ssRNA and have slightly different cell-type expression. TLR7 is expressed primarily in B cells, dendritic cells (DCs), and monocytes, whereas TLR8 is expressed primarily in DCs, monocytes, and granulocytes [103]. HERV-K RNA (and to some extent, HERV-W) has been linked to various neurological and central nervous system (CNS) disorders and is detectable in the blood, cerebrospinal fluid (CSF), and brain tissue of patients [104,105,106,107]. However, it was unknown whether this extracellular HERV-K RNA caused a phenotypic effect and which signaling pathway(s) it may activate. To identify the consequences of HERV-K (HML-2) RNA on TLR signaling, Dembny et al. investigated the effects of extracellular HERV-K (HML-2) RNA treatment on HEK293 cells that stably expressed a secretable alkaline phosphatase reporter gene under the control of an NF-κB-associated response element (to measure NF-κB signaling) in conjunction with either murine or human TLR7 or TLR8 (to measure if HML-2 RNA agonizes any of these TLRs) [108]. Interestingly, mTLR7 and hTLR8 responded to HERV-K (HML-2) RNA treatment and induced expression of the reporter gene, but not mTLR8 or hTLR7. Upon further analysis, they identified a sequence motif within the *env* gene of a subset of HML-2 proviruses (GUUGUGU) that is highly similar to the 20-nucelotide ssRNA40 derived from HIV that is known to act as a ligand for TLR7 and TLR8 [109,110]. A synthetic 22-nucleotide construct containing the GUUGUGU motif from the HML-2 *env* gene was shown to significantly induce NF-κB activation in human THP-1-derived macrophages, suggesting that this motif was responsible for the HML-2:hTLR8 interaction.

Although TLR7 and TLR8 are primarily expressed in immune cells, they are detectable in murine and human neurons as well [108]. Since the response to HML-2 RNA was similar between mTLR7 and hTLR8, mouse microglia and neurons were used as a model of how human microglia and neurons may react to extracellular HML-2 RNA. Interestingly, Dembny et al. found that while murine microglia responded to HML-2 RNA by inducing an inflammatory response, murine neurons underwent apoptosis. The differing outcome was suggested to be due to the different downstream signaling pathways that are activated depending on the cell type. Contrary to the microglia response, neurons did not induce NF-κB activation but did activate sterile alpha and TIR domain-containing 1 (SARM1), a member of the TIR domain-containing adapter protein family that mediates axonal degeneration [78]. Using the human neuronal cell line SH-SY5Y, the authors confirmed that HML-2 RNA induced neuronal death in humans and that upon neuronal cell death, they secrete their own HML-2 RNA that can then trigger apoptosis in exposed neurons. Overall, the authors found that HML-2 RNA is released upon neuronal death, thereby promoting further neural toxicity through (1) additional HML-2 RNA secretion and subsequent neuron apoptosis and (2) HML-2 RNA induced microglial pro-inflammatory activation, suggesting a prolonged pathogenic effect of HML-2 expression.

TLR9 is expressed in immune cells and was traditionally known to detect unmethylated CpG DNA. Recently, this view has shifted and TLR9 was shown to interact with RNA:DNA hybrids from virus-derived sequences to induce an anti-viral response through downstream Myd88 signaling [111]. As previously described, retroviruses reverse transcribe their RNA genome into DNA, which involves an RNA:DNA intermediate heterodimer. While it has not been demonstrated, it has been suggested that reverse transcribed HERV RNA may serve as an agonist for TLR9 signaling [79,80]. In a mouse model, TLR9 was shown to play a role in suppressing ERV expression and in modulating the anti-ERV antibody response [80]. This suggested that TLR9 actively senses the level of ERV expression and acts to limit their expression to a certain degree, through an unidentified mechanism; however, knockout of three TLRs (TLRs 3, 7, and 9) was required to see a significant effect on ERV activity, indicating that TLR9′s role in this was not essential and mostly likely redundant. Nonetheless, theses data suggest that TLR9 interacts with ERV products to some degree, with other work required to fully elucidate their relationship.

### 5.2. RIG-I-Like Receptors and MAVS Signaling

The retinoic acid-inducible gene (RIG)-I-like receptors (RLRs) are a ubiquitously expressed family of three PRRs: RIG-I (also known as DDX58), melanoma differentiation- associated protein 5 (MDA5, also known as IFIH1), and laboratory of genetics and physiology 2 (LGP2, also known as DHX58) [112]. Unlike the toll-like receptors (TLRs), which are embedded within membranes and restricted to surveying the cell surface and endosomal compartments, the RLRs are free-floating in the cytosol. All three RLRs bind to viral RNA via interactions between their central helicase domain and their carboxy-terminal domain (CTD), but only RIG-I and MDA5 contain the two N-terminal tandem caspase recruitment domains (2CARD) that are required for signal transduction. Instead, LGP2 is largely thought to serve as a regulator of RIG-I and MDA5 activity and downstream signaling [113,114,115,116].

Upon binding to viral RNA, RIG-I or MDA5 undergo a conformational change that exposes their doubled CARD domains [112]. This promotes RIG-I or MDA5 oligomerization and association with the CARD domain of the essential adapter protein, mitochondrial anti-viral signaling protein (MAVS), located on the mitochondrial surface [117,118,119]. This CARD–CARD domain interaction between RIG-I or MDA5 and MAVS leads to MAVS aggregation and the recruitment of downstream signaling molecules that eventually leads to TANK-binding kinase 1 (TBK1) and IκB kinase-ε (IKKε) activation. This ultimately results in the activation of the transcription factors IRFs 3/7 and NF-κB, thus promoting an anti-viral state and inducing the expression of IFN-I and other inflammatory cytokines [120,121].

While the RLRs are considered essential receptors for activating the innate immune response during viral infection, including influenza A, hepatitis C, vesicular stomatitis, encephalomyocarditis, and many others, relatively little work has focused on their ability to detect endogenous sources of viral-like RNA [122,123,124,125]. However, in spite of endogenization and extensive co-evolution of ERVs with their hosts, several independent research groups showed that HERV RNA can both bind to and activate RLRs for signal transduction. In our recent study, we exposed THP-1 human monocytic cells to ionizing radiation and saw a dramatic increase in the expression of various HERV families, especially the HERV-K (HML-2) subfamily [75]. Following pull-down of MDA5 and RT-qPCR analysis of HML-2 *env* in both the *sense* and *anti-sense* directions, we found that relatively equal amounts of sense and anti-sense HERV RNA was bound to MDA5. In the previously described study by Chiapinelli et al. that examined the interaction between HERV dsRNA and TLR3, the authors also found that HERV de-repression by DNMTi treatment leads to the activation of the MAVS pathway and the subsequent upregulation of ISGs. While it was not directly shown that the HERV RNA bound to either RIG-I or MDA5 to activate MAVS, it was suggested that at least MDA5 may respond to HERV RNA [36].

Other studies have also demonstrated a correlation between HERV upregulation, MAVS activation, and ISG induction. KAP1 associates with krüppel-associated box domain zinc-finger proteins (KZNFs) to mediate cytosine methylation of HERVs. Through shRNA-mediated knockdown and CRISPR-mediated knockout, Tie et al. demonstrated that KAP1 suppresses HERV expression in multiple cell types, including PBMCs, NTERA-2 cells, HeLa, and 293T cells [126]. The induction of a specific HERV group, HERVK14C, was shown to be upregulated multiple folds higher than other types of retroelements, such as LIPA4 (L1 subfamily) and SVA D (SVA subfamily), in response to KAP1 knockdown and knockout in each cell line examined. Other HERVs, including HERV18, LTR6B, and HERVS71, were also identified to be upregulated in response to KAP1-KO. Interestingly, KAP1 knockdown resulted in the upregulation of various ISGs (ISG56, CCL5, CXCL10, etc.) and this effect was demonstrated to be mediated through MAVS signaling [126]. However, the precise mechanism of HERV-mediated MAVS activation was not elucidated.

A similar study by Sun et al. demonstrated that at least twelve ERVs are induced during mesothelioma development in a mouse model, leading to increased dsRNA levels and activation of a type I IFN response [127]. Similar to the mitigation of KAP1-KO-induced ISG expression, siRNA against MAVS was able to reduce the basal ISG signature.

Together, these studies suggest that HERV binds to MDA5 (and potentially RIG-I) to activate MAVS, which subsequently induces type I IFN and ISG expression; however, more research is required to definitively prove this. Nonetheless, there is a clear association between high HERV expression and innate immunity-related gene induction, which can be reduced through inhibiting MAVS signaling.

### 5.3. cGAS/STING Pathway

The detection of foreign DNA is also an important element of innate immunity and is mediated in part by cGAS [71,72]. Unlike the other PRRs, cGAS induces downstream signaling through the synthesis of a secondary messenger molecule. Upon recognition of dsDNA, cGAS undergoes a conformational change that results in the activation of its 2′3′ cyclic GMP-AMP (cGAMP) producing ability. The resulting cGAMP then serves as an activator of stimulator of interferon gene (STING). Similar to the role of MAVS in RLR signaling, STING-mediated activation of TBK1 and TRAF6 (which subsequently activates IKKε) results in the activation of the transcription factors IRF 3/7 and NF-κB, respectively [71,72].

HERVs have the potential to form dsDNA products following expression and reverse transcription, suggesting that they may serve as agonists of cGAS-STING signaling. Lima-Junior et al. recently performed an extensive series of experiments in mice that demonstrated that reverse transcribed ERVs may be critical in promoting homeostatic and inflammatory responses to microbiota in the skin through cGAS-STING pathway activation. [81]. Specifically, in response to commensal bacteria colonization of the skin, ERVs are upregulated in keratinocytes and correlated with a type I IFN response. To prove that reverse transcription was involved, the authors treated mice with tenofovir and emtricitabine, two commonly used reverse transcriptase inhibitors, prior to bacteria colonization. This resulted in a blunted T cell response, including reduced bacteria-specific CD8+ T cells and a lower overall abundance of all T cell subtypes within the colonized skin while secondary lymphoid organs were unaffected. The keratinocytes in the colonized skin of the anti-retroviral treated mice displayed a similar abundance of ERV expression as the non-treated counterparts; however, single-cell sequencing revealed a stark discrepancy in the expression of genes associated with anti-viral/anti-microbial defense and wound healing. Knockout of cGAS or STING led to similar results as the administration of RTi treatment, with a significant decrease in T cell activity and innate immunity-related gene expression in the knockout mice compared to WT mice following skin colonization. Overall, the authors suggest that ERVs contribute to our response to microbiota through regulating the activation of our innate responses [81].

## 6. HERVs in a Clinical Setting

When studying the link between HERV expression and the presence of a disease state, it is a challenging task to determine whether (1) HERV upregulation is a preceding event that facilitated the development of the disease state, (2) HERV upregulation is an active byproduct of the disease state that contributes to the maintenance or worsening of the disease state, or (3) HERV upregulation is a non-participating byproduct of the disease state without an active function. While many of the following studies only provide a correlation between HERV expression and the presence of a disease, and therefore do not directly answer the question about the role of HERVs in the context of that disease, the previous portions of this review that focused on the mechanisms relating HERV activity to the innate immune response and the augmentation of pro-inflammatory gene expression suggest the potential involvement of HERV expression in pathogenesis of the diseases. The examples below are just a few of many instances where HERV expression can be tied to chronic inflammation and autoimmunity.

### 6.1. HERVs and Chronic Inflammation/Autoimmunity: Multiple Sclerosis

MS is a chronic autoimmune disease characterized by the immune system attacking the protective sheath (myelin) that covers and protects nerve fibers in the central nervous system (CNS). The precise cause of MS is unknown, but it is speculated to arise from an environmental event, such as viral infection, that can trigger an autoimmune response in genetically predisposed individuals [128]. For instance, people who experience Epstein–Barr virus infection are a reportedly 32 times more likely to develop MS than non-infected counterparts [129]. No other known risk factor (such as HLA-DR15 allele homozygosity) had as strong of a correlation with MS as EBV infection and other viruses did not show a significant correlation with MS, including cytomegalovirus (CMV), a herpesvirus that is transmitted through the saliva, similar to EBV. As previously discussed, MS patients typically express high levels of MSRV and other HERVs belonging to the HERV-W family. Specifically, there is a 1.5- to 3-fold increase in HERV-W Env expression, which acts as a TLR4:CD14 agonist and induces the expression of various pro-inflammatory cytokines to promote neuroinflammation [130].

Interestingly, it was found that EBV glycoprotein 350 (EBVgp350), the major envelope protein of EBV, can stimulate the expression of HERV-W in astrocytes, monocytes, and B cells, likely through the NF-κB pathway [131]. This could explain the link between EBV, HERV-W, and the development of MS; wherein, EBV infection stimulates HERV-W expression, which in turn, induces a pro-inflammatory response through TLR4 activation and subsequently drives autoimmunity. This “dual virus hypothesis” relies on the presence of EBV infection shortly before the development of MS. However, while MS patients are significantly likely to have had an EBV infection at one point in time, there is not a direct link between active EBV infection and MS development. It may be possible that once activated by EBV, HERV-W does not rely on EBV infection for its maintenance and may perpetuate its upregulation through a gradual increase in neuroinflammation and degeneration. This is supported by data that show a return of HERV-W expression to baseline following successful intervention (determined by a reduced inflammatory signature and neurodegeneration) with IFNβ therapy, suggesting that HERV-W expression relies on a consistent supply of inflammation [132].

While HERV-W Env antigen is detectable in the serum of 73% of MS patients (and 0% of healthy controls), it does not appear that HERV-W antigen levels are directly correlated to the severity of the symptoms [72]. However, from the same study, the HERV-W DNA copy number in PBMCs is increased in chronic progressive MS patients versus relapsing-remitting MS (RRMS) patients and healthy controls, suggesting that HERV-W proviruses undergo expression and re-integration events during active MS. A separate study using an RNA-sequencing approach to more acutely detect the presence and expression levels of specific HERV loci in MS patients revealed that patients with a higher Expanded Disability Status Scale (EDSS) score (indicating a more severe MS condition) have a higher number of active HERV-W loci compared to relapsing and low score EDSS patients [133]. Interestingly, they also identified 18 other families of HERVs that were more highly expressed in MS patients compared to healthy controls. This suggests that the activation of HERV-W may contribute to the secondary activation of numerous groups of HERVs through the mechanisms described above.

Based on these findings, a monoclonal antibody (GNbAC1) against HERV-W Env was developed for the treatment of MS and showed no signs of toxicity in early clinical testing. In 2021, it completed a randomized phase 2b study with 270 participants (clinical trial: NCT04480307). Although the primary endpoint was not met (reduction in the cumulative number of T1 gadolinium-enhancing (GdE) lesions) at 24 weeks, the monoclonal antibody was found to exert anti-neurodegenerative effects based on other criteria, such as measures of brain atrophy, myelin integrity, and the number of chronic T1 hypointense lesions [134]. Following these findings, a new phase 2 study will be conducted with an extended dose range (instead of doses ranging from 6–18 mg/kg, doses will range from 18–54 mg/kg) to see if higher concentrations can elicit a significant response in the endpoint criteria.

### 6.2. Rheumatoid Arthritis

Rheumatoid arthritis (RA) is a chronic autoimmune disease that is associated with autoantibodies against various self-epitopes present in the joints. Interestingly, it is speculated that upregulation of HERV expression, and specifically translation, may act as a trigger for the development of autoantibodies. Evidence suggests that RA patients have cross-reactive antibodies towards HERV-derived antigens with regions of high structural homology to “host” antigens (proteins of non-viral origin) present in the joints [135,136]. For instance, HERV-K10 Gag1 protein shares an antigenic region with Collagen II, which is highly expressed in joints, and Freimanis et al. demonstrated that human fibroblast-like synoviocytes derived from a RA patient displayed several fold higher expression of HERV-K10 *gag1* compared to samples from osteoarthritis (OA) and healthy donors [135]. The increase in HML-2 expression was confirmed by Reynier et al., who demonstrated that both types of HML-2 (HML-2 type I and type II) are significantly upregulated in the synovial fluid of RA patients compared to OA patients and healthy controls, with a higher HML-2 type I viral load associated with increased disease activity in RA patients [136]. The increased expression of HML-2 in RA patients found by Freimanis et al. was also reflected by an increased titer of autoantibodies against a recombinant peptide derived from the antigenic region of HERV-K10 Gag1 that shares homology with Collagen II, suggesting a direct link between elevated HML-2 expression, anti-HML-2 autoantibodies, and anti-Collagen II autoantibodies [135]. A similar study by Nelson et al. also demonstrated a significantly higher titer of anti-HERV-K10 antibodies in RA patients compared to controls [137]. However, cross-reactivity of the anti-HML-2 antibodies against Collagen II was not performed by Freimanis et al. beyond the initial bioinformatic analysis to identify homologous regions. Additionally, it is unknown if the elevated HML-2 expression and autoantibodies occurs prior to or following the development of RA, as pro-inflammatory cytokine treatment of fibroblast-like synoviocytes induces the expression of HML-2 and therefore implies that HML-2 activity could simply be the result of RA. Therefore, future studies on the precise timing of HML-2 upregulation in RA patients are needed to address this question.

### 6.3. Systemic Lupus Erythematosus

Systemic lupus erythematosus (SLE) is a systemic autoimmune disease that is characterized by a dysregulation in both the innate and adaptive immune systems, leading to the development of autoantibodies, typically those against nuclear components [138]. These antinuclear antibodies (ANAs) bind to DNA, RNA, and nucleic acid:protein complexes and exacerbate pro-inflammatory signaling, tissue infiltration by immune cells, and ultimately, tissue destruction [139,140]. Due to previous associations between HERV expression and SLE, Tokuyama et al. implemented RNA-sequencing on PBMCs derived from SLE patients and healthy controls to examine HERV expression on a locus-specific level using the software package they developed, ERVmap [141,142]. They identified over 100 unique ERV loci that are overexpressed in SLE patients and found that the total ERV read count significantly correlated with measures of disease severity, such as anti-nuclear, anti-double stranded DNA, anti-ribonucleoprotein (anti-RNP), and anti-Smith (Sm) anti-bodies. A separate study using the software package Telescope to characterize locus-specific HERV expression was released around the same time and confirmed these results, with over 300 loci significantly upregulated in SLE patients compared to controls in contrast to 10 downregulated loci [138]. Following these findings, the Tokuyama group analyzed the HML-2 subfamily due to its high degree of coding competence compared to other families and found that 4 of the 12 HML-2 loci (HERV-K102, -K106, -K110, and -K115) with envelope-coding sequences are significantly upregulated in SLE PBMCs [141]. Interestingly, these loci appear to be human-specific, with no known homology to other primate genomes. Of the four loci, HERV-K102 displayed significant correlation with anti-RNP titers. Based on this, Tokuyama et al. produced a recombinant envelope SU peptide based on HERV-K102 and assayed for autoantibodies against it in SLE patients and found that anti-HERV-K102 antibody titers correlated with the ISG signature of circulating PBMCs [141].

Following this finding, Tokuyama et al. moved to another aspect of innate immunity, neutrophils. A key driver of inflammation in SLE is neutrophil activity and the formation of neutrophil extracellular traps (NETs), an inflammatory marker of SLE that some speculate may be involved in the development of ANAs, and subsequently, SLE [143,144,145]. To examine if HERV-K102 Env-SU can activate neutrophils, Tokuyama et al. incubated recombinant Env-SU antigen in plasma from SLE patients or healthy controls. The immune complexes generated in the plasma from SLE patients, but not healthy controls, were able to activate neutrophil phagocytosis and induce the secretion of intracellular DNA in the form of NETs [141]. Overall, the authors suggest that HERV-K102 antigen and anti-HERV-K102 IgG from SLE patients form immune complexes that are readily phagocytosed by neutrophils and induce NET formation, which may contribute to autoimmunity and enhanced interferon signaling in SLE.

### 6.4. Pulmonary Arterial Hypertension

Pulmonary arterial hypertension (PAH) is a progressive disorder that is characterized by endothelial dysfunction and vasculature remodeling of the pulmonary arteries, leading to obstruction of the blood flow and increased resistance [146,147]. The increase in blood vessel wall resistance causes an elevation in pulmonary artery pressure that can overload the right ventricular, resulting in heart failure and death. Although PAH is known to correlate with various genetic and environmental factors, including mutations in the BMPR2 gene, the presence of pre-existing connect tissue diseases (SLE, rheumatoid arthritis, systemic sclerosis, etc.), HIV/schistosomiasis infection, and others [148], the majority of PAH cases are idiopathic (occur without a known cause) [149]. Recent research suggests that inflammation and autoimmunity are intrinsically linked to the development with PAH, with Saito et al. providing evidence of HERV-K as a novel initiator and sustainer of PAH that may be a suitable target for therapeutic intervention [150,151,152].

To determine if lung tissue from PAH patients contained viruses that were previously implicated in PAH pathology (HIV, Human herpesvirus 8, and hepatitis C virus), Saito et al. performed a metagenomic viral screen using next-generation sequencing and did not detect the presence of any exogenous viruses [152]. However, the authors observed a significant increase in HERV-K envelope and dUTPase mRNA in lung extracts from patients with PAH compared to healthy controls. Upon further inspection, they identified that HERV-K envelope and dUTPase proteins were primarily expressed in CD68+ macrophages that were present in the lung tissue and found that circulating monocytes from PAH patients exhibited higher levels of dUTPase mRNA compared to controls. These findings led the authors to explore the functional implications of dUTPase elevation by treating monocytes, pulmonary arterial endothelial cells (PAECs), and B cells with recombinant dUTPase. Following dUTPase treatment, all three cell types exhibited markers of inflammation or activation, including secretion of TNFα, IL1β, and IL6 by the monocytes and PAECs, and with CD69 expression and STAT3 signaling in the B cells. Additionally, the authors found that HERV-K dUTPase treated PAECs displayed increased vulnerability to apoptosis in an IL-6 independent manner in response to serum withdrawal, a hallmark of PAH PAECs. Lastly, as the authors found increased HERV-K dUTPase levels in the lungs and circulating monocytes of PAH patients, they decided to examine the impact of intravenous HERV-K dUTPase administration in a rat model to assess if HERV-K dUTPase elevation could initiate the development of PAH. They found that HERV-K dUTPase administration resulted in decreased pulmonary artery acceleration time, increased right ventricular systolic blood pressure, and right ventricular hypertrophy compared with saline-injected rat markers of early PAH development. Interestingly, the same group recently showed that monocytes overexpressing HERV-K dUTPase release this protein, incorporated into extracellular vesicles (EVs), and cause pulmonary hypertension in association with endothelial mesenchymal transition (EndMT) related to the induction of SNAIL/SLUG, IL-6, and VCAM1 [153].

While it is difficult to definitively prove that HERV-K elevation is the (or an) initiating factor in PAH, the authors identified that HERV-K dUTPase is elevated in PAH patients, demonstrated the ability of HERV-K dUTPase to induce inflammation through the activation of several relevant cell types, and established that HERV-K dUTPase administration alone can result in pathological indicators of PAH development [152]. Though there is a relative lack of literature on the relationship between HERVs and cardiovascular-related diseases, it is clear that HERV-K elevation can instigate and sustain inflammation within the vasculature network. It is likely that future studies can shed new light on the consequences of HERV reactivation in the context of heart inflammation and disease.

## 7. Concluding Remarks

In the present review, we detailed the molecular mechanisms involved with the regulation and reactivation of HERV expression following changes in the epigenetic landscape and in response to various environmental stimuli. Subsequently, we explored the ability of HERV-derived products, including ssRNA, dsRNA, RNA:DNA duplexes, dsDNA, proteins, and RNPs of retroviral origin, to act as viral-like agonists in the activation of PRR signaling. Finally, we discussed the clinical association between activation of the HERV virome and the prevalence or severity of several chronic inflammatory diseases, with data suggesting that HERVs may play an active role in the exacerbation of pro-inflammatory signaling in these disease states.

Due to the success of high-throughput sequencing methods for genomic and transcriptomic analyses, our knowledge about the involvement of endogenous LTR retroelements in the innate immune response is steadily increasing. The existing data allow us to theorize that during the course of evolution, at least in mammals, LTR retroelements were co-opted by the immune system and incorporated as intrinsic enhancers of the innate immune response. Potentially, HERV-derived products could act as transmitters of PRR-mediated signaling cascades to regulate the inflammatory state of tissues or organs, such as through the activation of anti-viral RNA- or DNA-sensing pathways that lead to type I interferon expression (Figure 5). In such a situation, HERVs would be induced by environmental stimuli or cellular stress, resulting in a pro-inflammatory response that could then act in both an autocrine and paracrine manner to perpetuate the upregulation of HERV expression and the propagation of inflammation-related activities. Of note, this ‘commensalism’ requires a mechanism to silence or inhibit HERV activity following the resolution of the initial stimuli (such as downregulation of HERV expression following the removal of all pathogenic material from a previously infected tissue) in order to prevent detrimental hyperactivation of the immune response. Otherwise, there would be constant activation of HERV expression via interferon signaling feedback loops and other pro-inflammatory cytokine activities. The mechanism(s) that underlie the mediation of HERV activation and repression remain to be fully investigated. To that end, our lab continues to explore the transcriptional activation of distinct HERV proviral loci and their role in reinforcing specific PRR signaling pathways in the context of normal and pathogenic innate immune responses.

In summary, a wealth of literature supports the notion of HERVs as essential components of innate immunity. However, a greater understanding of the mechanisms behind HERV-mediated modulation of the innate immune response is required and may improve our basic understanding of the regulation of normal and abnormal immune activation. Ultimately, the continuation of HERV-based research projects may lead to the identification of new therapeutic targets to counteract local and systemic inflammation.

## Figures and Tables

**Figure 1 pathogens-12-00162-f001:**
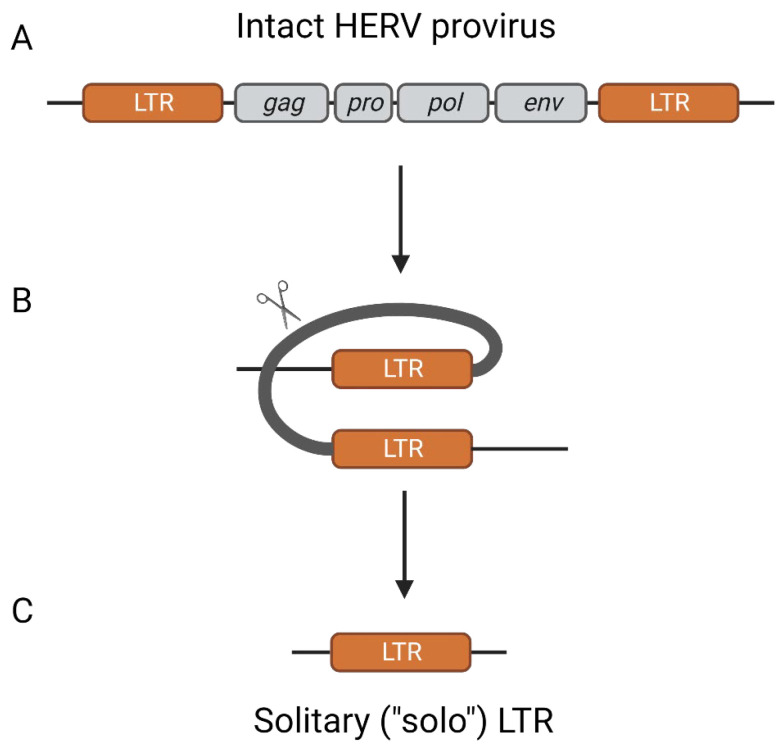
HERV provirus structure and the formation of a solo LTR. The structure of a typical HERV provirus includes gag, pro, pol, and env encoding regions with flanking identical LTRs (**A**). Due to the homology between the flanking LTRs, they can undergo homologous recombination (**B**) and form solitary (“solo”) LTRs (**C**) [8,9].

**Figure 2 pathogens-12-00162-f002:**
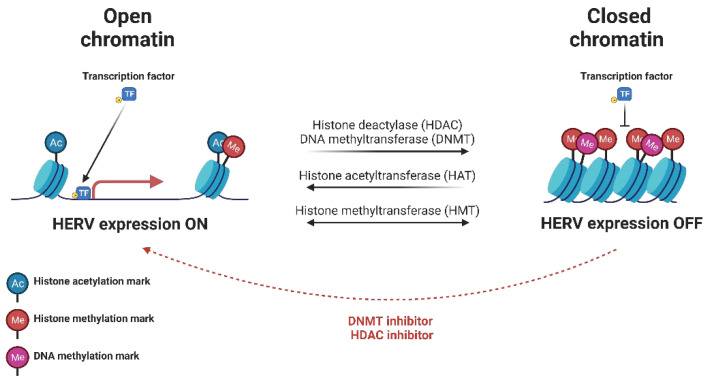
Epigenetic regulation of HERV expression. The abundance of DNA methylation, histone methylation, and histone acetylation influences the strength of the interaction between DNA and the surrounding histone, leading to either an “open” or “closed” chromatin state. Open chromatin is permissive to HERV expression and allows for its upregulation if the required transcription factor(s) are readily available. Closed chromatin blocks HERV expression by reducing the accessibility of the HERV to transcription factors and other transcription-related machinery (citations in text).

**Figure 3 pathogens-12-00162-f003:**
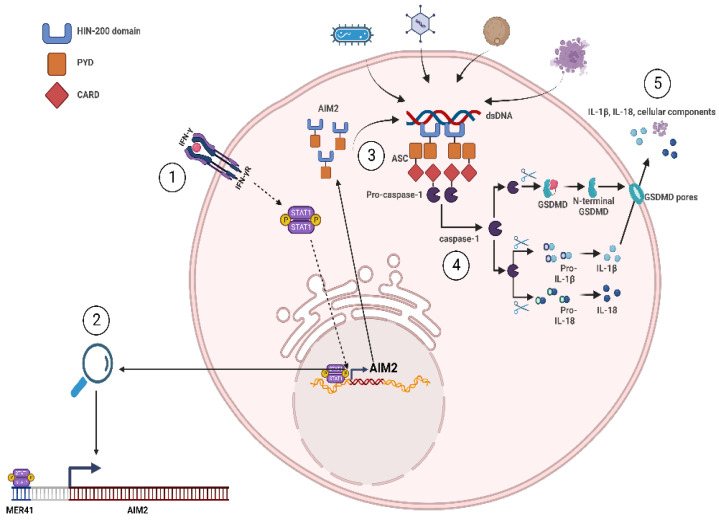
HERV solo LTRs as enhancers of innate immunity. Following interferon gamma signaling (1), STAT1 homodimers bind to the MER41 element upstream of AIM2 (termed MER41.AIM2) to induce AIM2 expression (2). This primes the cell for pathogen-derived and self-derived dsDNA sensing by the HIN-200 domain of AIM2, which leads to the formation of the AIM2 inflammasome, consisting of AIM2, ASC, and pro-caspase-1 (3). Following, pro-caspase-1 undergoes proteolytic cleavage into caspase-1 and cleaves GSDMD, pro-IL-1B, and pro-IL-18 into their active forms (4). This results in GSDMD pore formation and the secretion of IL-1B, IL-18, and cellular components related to pyroptosis in order to stimulate a robust pro-inflammatory and innate immune response (5) [66,68,69].

**Figure 4 pathogens-12-00162-f004:**
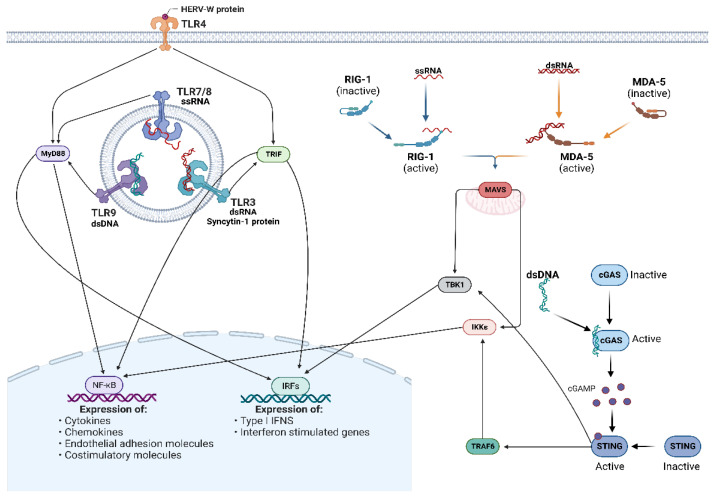
HERVs as activators of pattern recognition receptors and PRR-mediated pathways. HERV-derived DNA, RNA, and protein can be sensed by three distinct types of PRRs (TLRs, RLRs, and cGAS). Following PRR stimulation, different downstream signaling pathways are activated and result in the induction of an innate immune response through the transcription factors NF-kB and IRFs 3/7. The activation of these transcription factors results in an anti-viral state that mimics the effects of an exogenous infection, leading some to describe the actions of HERVs as *viral mimicry* (citations in text).

**Figure 5 pathogens-12-00162-f005:**
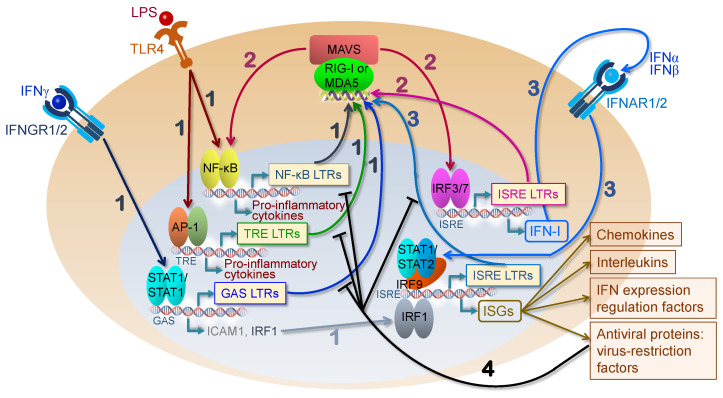
Schematic diagram depicting involvement of HERV expression in regulation of PRR-mediated type I interferon expression and signaling (shown for RLR-MAVS pathway) and potential feedback loops that affect the transcriptional activity of different proviral loci. Interferon gamma and LPS, potent activators of innate immune response, upregulate transcription of different HERV loci, depending on the specific TFBSs in their own promoters or the promoters of their host genes. Viral RNA binding receptors (RIG-I/MDA5 in the diagram) trigger downstream signaling that results in IFN-I expression (1). RNA sensor-induced signaling results in the activation of IRF3/7- and NF-κB-dependent transcription and thereby upregulates the transcription of HERV loci that contain ISRE and NF-κB TFBS, whose RNA may in turn activate the RNA sensors (2). Since certain HERVs contain STAT1/2 binding motifs in their promoters, RLR signaling may be recurrently activated via IFN-I-mediated HERV transcription in a positive feedback mechanism (3). Since many ISGs are restriction factors of viral replication, their expression can inhibit HERVs, providing negative feedback (4). NF-κB LTRs, TRE LTRs, GAS LTRs, and ISRE LTRs refer to HERV loci whose LTR promoters contain NF-κB, AP-1, STAT1/1, and STAT1/2 transcription factor binding sites, respectively.

**Table 1 pathogens-12-00162-t001:** Pattern recognition receptors shown to interact with different HERV-derived products.

PRR	Type of Antigen	Source
**TLR3**	HERV-derived dsRNA; syncytin-1 protein	[36,75,76]
**TLR4**	HERV-W Env	[77]
**TLR8**	HERV-K (HML-2) RNA	[78]
**TLR9**	HERV-derived RNA:DNA heteroduplexes (speculative)	[79,80]
**MDA5**	HERV-derived dsRNA	[36,75]
**cGAS**	HERV-derived dsDNA	[81]

## Data Availability

Not applicable.

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
