# Peer review of "Endogenous Retroviruses as Modulators of Innate Immunity"

_pathogens, 2023, doi:10.3390/pathogens12020162_

Round 1
Reviewer 1 Report
This is a very nice review covering ER, their gene regulation and interaction and induction with components of the cellular innate immune system.
Minor comments:
- The position of Fig. 2 is too early and should not before its first mentioning.
- Line 316: References missing
- Section 5, line 318 ff: the authors use the term “viral mimicry” but do not explain this term; please discuss what viral mimicry means.
- Line 324: in the list of PRR, cGAS/STING should be included
- The position of table 1 should earlier, around line 333
Author Response
We would like to thank the reviewers for the significant comments and suggestions that helped us to include additional important information, improve figures and overall strengthen the manuscript.
Below is our point-by-point response to the reviewer’s comments. The reviewer’s comments are numbered and in italic; our responses are in plain font, with changes to the written text of the review paper underlined.
Reviewer #1:
- The position of Fig. 2 is too early and should not before its first mentioning.
- We are thankful for this suggestion, we followed it and moved the figure as suggested.
- Line 316: References missing
- This is a good catch. We added the missing reference.
- Section 5, line 318 ff: the authors use the term “viral mimicry” but do not explain this term; please discuss what viral mimicry means.
-We appreciate this valuable criticism, we can see how it is important to clarify what “viral mimicry” is referring to, since it is not exactly straight forward. It was originally used by a paper published in 2015 to describe how HERV reactivation “tricks” the cell into thinking it is infected (by activating similar PRRs), leading to an antiviral immune response that mimics the response to exogenous viruses. This is why they termed it “viral mimicry.” Shown below is the added context to the paper:
“While not all of these receptor classes have demonstrated the ability to recognize HERV-derived products, the viral-like nature of particular HERV-derived DNA, RNA, and proteins allows them to be recognized by and stimulate various PRRs to induce an innate immune response, similar to that of exogenous viruses. As a result, the effects of aberrant HERV upregulation are said to mimic the antiviral state that is induced following viral infection, leading to coinage of the term “viral mimicry” when discussing the implications of HERV reactivation on the innate immune response [40, 74, 75].”
For reference, viral mimicry was first mentioned in this paper:
Roulois, D., et al., DNA-Demethylating Agents Target Colorectal Cancer Cells by Inducing Viral Mimicry by Endogenous Transcripts. Cell, 2015. 162(5): p. 961-973.
- Line 324: in the list of PRR, cGAS/STING should be included
- This is a very useful suggestion and we followed it. Shown below is the added text:
“Most PRRs in vertebrate biology can be groups into five major classes: I) toll-like receptors, II) retinoic-acid inducible (RIG)-I-like receptors (RLRs), III) C-type lectin receptors (CLRs), IV) nucleotide oligomerization domain (NOD)-like receptors (NLRs), and V) absent in melanoma-2 (AIM2)-like receptors (ALRs) [71]. An important PRR which does not fall into these five major groups is cyclic GMP–AMP synthase (cGAS), due to its relatively unique mechanism of inducing downstream signaling (as discussed later) [72, 73].”
- The position of table 1 should earlier, around line 333
- This suggestion is valuable and we followed it by moving the table to the suggested location (due to other changes in the review paper, it is not line 333 anymore, but the general location of the figure is in the location that the reviewer recommended).
Reviewer 2 Report
Eric Russ and Sergey Iordanskiy have done wonderful job by writing this essential review article to emphasize the study of HERV and various human disorders. However, following concerns may be addressed to improve the MS further:
Please mention the full form of pro inflamm cytokines in Fig. 3 legend.
Section 5 should contain a descriptive figure.
Section 3 should contain a descriptive figure as well.
Inflammation is one of the very common causes of cardiovascular disorders. However, entire manuscript has no mention about that disease and its link with HERV. Please write a section/para for the same.
A table may be added in the section 2, to summarize all retroviruses emphasized in the current review article.
Author Response
We would like to thank the reviewers for the significant comments and suggestions that helped us to include additional important information, improve figures and overall strengthen the manuscript.
Below is our point-by-point response to the reviewer’s comments. The reviewer’s comments are numbered and in italic; our responses are in plain font, with changes to the written text of the review paper underlined.
Reviewer #2:
- Please mention the full form of pro inflamm cytokines in Fig. 3 legend.
- We are thankful for this suggestion, we followed it and changed “Pro-inflamm” to “Pro-inflammatory”.
- Section 5 should contain a descriptive figure.
- This is a valuable suggestion that strengthens the review, we followed it and added a figure (Figure 4).
- Section 3 should contain a descriptive figure as well.
- This is a valuable suggestion, we followed it and added a figure (Figure 2).
- Inflammation is one of the very common causes of cardiovascular disorders. However, entire manuscript has no mention about that disease and its link with HERV. Please write a section/para for the same.
- This is a very valid point. Surprisingly, there is not as a much literature for HERVs in relation to cardiovascular diseases, but there is relatively convincing evidence that HERV-K may be implicated in pulmonary arterial hypertension. Shown below is what we added to the paper:
“Pulmonary arterial hypertension
Pulmonary arterial hypertension (PAH) is progressive disorder that is characterized by endothelial dysfunction and vasculature remodeling of the pulmonary arteries, leading to obstruction of the blood flow and increased resistance [147, 148]. The increase in blood vessel wall resistance causes an elevation in pulmonary artery pressure that can overload the right ventricular, resulting in heart failure and death. Although PAH is known to correlate with various genetic and environmental factors, including mutations in the BMPR2 gene, the presence of pre-existing connect tissue diseases (SLE, rheumatoid arthritis, systemic sclerosis, etc.), HIV/schistosomiasis infection, and others [149], the majority of PAH cases are idiopathic (occur without a known cause) [150]. Recent research suggests that inflammation and autoimmunity are intrinsically linked to the development with PAH, with Saito et al. providing evidence of HERV-K as a novel initiator and sustainer of PAH that may be a suitable target for therapeutic intervention [151-153].
To determine if lung tissue from PAH patients contained viruses that were previously implicated in PAH pathology (HIV, Human herpesvirus 8, and hepatitis C virus), Saito et al. performed a metagenomic viral screen using next-generation sequencing and did not detect the presence of any exogenous viruses [153]. However, the authors observed a significant increase in HERV-K envelope and dUTPase mRNA in lung extracts from patients with PAH compared to healthy controls. Upon further inspection, they identified that HERV-K envelope and dUTPase proteins were primarily expressed in CD68+ macrophages that were present in the lung tissue and found that circulating monocytes from PAH patients exhibited higher levels of dUTPase mRNA compared to controls. These findings led the authors to explore the functional implications of dUTPase elevation by treating monocytes, pulmonary arterial endothelial cells (PAECs), and B cells with recombinant dUTPase. Following dUTPase treatment, all three cell types exhibited markers of inflammation or activation, including secretion of TNFα, IL1β, and IL6 by the monocytes and PAECs, and with CD69 expression and STAT3 signaling in the B cells. Additionally, the authors found that HERV-K dUTPase treated PAECs displayed increased vulnerability to apoptosis in an IL-6 independent manner in response to serum withdrawal, a hallmark of PAH PAECs. Lastly, as the authors found increased HERV-K dUTPase levels in the lungs and circulating monocytes of PAH patients, they decided to examine the impact of intravenous HERV-K dUTPase administration in a rat model to assess if HERV-K dUTPase elevation could initiate the development of PAH. They found that HERV-K dUTPase administration resulted in decreased pulmonary artery acceleration time, increased right ventricular systolic blood pressure, and right ventricular hypertrophy compared with saline-injected rats – markers of early PAH development. Interestingly, the same group recently showed that monocytes overexpressing HERV-K dUTPase release this protein, incorporated into extracellular vesicles (EVs), and cause pulmonary hypertension in association with endothelial mesenchymal transition (EndMT) related to the induction of SNAIL/SLUG, IL-6, and VCAM1 [154] .
While it is difficult to definitively prove that HERV-K elevation is the (or an) initiating factor in PAH, the authors identified that HERV-K dUTPase is elevated in PAH patients, demonstrated the ability of HERV-K dUTPase to induce inflammation through the activation of several relevant cell types, and established that HERV-K dUTPase administration alone can result in pathological indicators of PAH development [153]. Though there is a relative lack of literature on the relationship between HERVs and cardiovascular-related diseases, it is clear that HERV-K elevation can instigate and sustain inflammation within the vasculature network. It is likely that future studies can shed new light on the consequences of HERV reactivation in the context of heart inflammation and disease.”
- A table may be added in the section 2, to summarize all retroviruses emphasized in the current review article.
-This is a valuable suggestion. However, we attempted to make a table that summarizes all of the retroviruses from our review, but we ran into some technical challenges that prevented us from being able to make a neat and coherent table. We apologize for not meeting this suggestion.